# Structural basis for LIN54 recognition of CHR elements in cell cycle-regulated promoters

Aimee H. Marceau[1], Jessica G. Felthousen[2], Paul D. Goetsch[3], Audra N. Iness[2], Hsiau-Wei Lee[1], Sarvind M. Tripathi[1], Susan Strome[3], Larisa Litovchick[2] & Seth M. Rubin[1]

The MuvB complex recruits transcription factors to activate or repress genes with cell cycle-dependent expression patterns. MuvB contains the DNA-binding protein LIN54, which directs the complex to promoter cell cycle genes homology region (CHR) elements. Here we characterize the DNA-binding properties of LIN54 and describe the structural basis for recognition of a CHR sequence. We biochemically define the CHR consensus as TTYRAA and determine that two tandem cysteine rich regions are required for high-affinity DNA association. A crystal structure of the LIN54 DNA-binding domain in complex with a CHR sequence reveals that sequence specificity is conferred by two tyrosine residues, which insert into the minor groove of the DNA duplex. We demonstrate that this unique tyrosine-mediated DNA binding is necessary for MuvB recruitment to target promoters. Our results suggest a model in which MuvB binds near transcription start sites and plays a role in positioning downstream nucleosomes.

---

[1] Department of Chemistry and Biochemistry, University of California, 1156 High Street, Santa Cruz, California 95064, USA. [2] Division of Hematology, Oncology and Palliative Care and Massey Cancer Center, Virginia Commonwealth University, Richmond, Virginia 23298, USA. [3] Department of Molecular, Cell and Developmental Biology, University of California, Santa Cruz, California 95064, USA. Correspondence and requests for materials should be addressed to S.M.R. (email: srubin@ucsc.edu).

Control of the cell cycle is critical for the development of multicellular organisms and cell cycle deregulation is a hallmark of cancer. Transcription factors organize the spatial and temporal processes underlying cell cycle progression and exit, which are necessary for proliferation and differentiation[1–3]. One strategy for this regulation is the use of protein complexes that use different binding partners to either activate or inhibit transcription. The conserved MuvB complex acts as a switchable master cell cycle regulator, promoting expression of cell cycle genes during the cell cycle and maintaining repression of cell cycle genes during quiescence[2–5].

In mammalian cells, MuvB binds the retinoblastoma tumour suppressor protein paralogues, p130 or p107, and the transcription factor heterodimer, E2F4-DP, to repress expression of cell cycle genes in G0 and G1 (refs 6–8). This 'DREAM' complex promotes quiescence and is required for development[1,2,9–11]. Mice deficient in DREAM complex assembly have defects in bone development resulting in postnatal death[12]. When cells enter the cell cycle, cyclin-dependent kinase activity dissociates MuvB from the other subunits of DREAM and MuvB then associates with the B-Myb (Myb) transcription factor. This Myb–MuvB complex activates expression of a subset of late cell cycle genes and promotes activity of the FoxM1 transcription factor[1,2,4–6,8,10]. MuvB is important for Myb and FoxM1 activity by a mechanism that is not well understood but probably involves recruitment of these transcription factors to promoters.

MuvB contains five proteins: LIN54, LIN37, LIN52, LIN9 and RBAP48 (refs 1,9,10). Little is known about the biochemical functions of these proteins other than LIN52 binds p130 (ref. 6), LIN54 binds DNA[13] and RBAP48 binds histones when present in other complexes[11,14,15]. LIN54 is responsible for directing MuvB binding to the cell cycle genes homology region (CHR) of target genes[13,16]. The CHR element is located in the promoters of cell cycle-regulated genes and has been implicated in gene repression in G0 and early G1, and gene activation in G2 and M phases[16–18]. The most common CHR motif is 5′-TTTGAA-3′, although other sequence variants have been identified[16,18]. The mechanism by which LIN54 binds the CHR and the precise sequence limitations are unknown and is the focus of this study.

The DNA-binding domain (DBD) of LIN54 contains two tandem cysteine-rich (CXC) domains that share sequence similarity with tesmin, a testis-specific metallothionein-like protein[19,20]. The CXC domain consists of nine cysteines that enable LIN54 binding to DNA[19–24]. The structure of a CXC domain from MSL2, part of the *Drosophila* male-specific lethal dosage compensation complex, is known[23,24]. Although MSL2 contains a single CXC domain, the majority of proteins homologous to LIN54 and tesmin contain two CXC domains in tandem. In fact, the MSL2 DNA-binding residues are not conserved in the tesmin family.

We describe here the association of LIN54 with the CHR DNA element. We show that LIN54 uses both CXC domains to recognize the specific DNA consensus motif TTYRAA. We determined a 2.4 Å X-ray crystallography structure of LIN54 bound to a 13 bp DNA with a centrally located CHR. The structure reveals that critical tyrosines, one from each CXC domain, specifically recognize the TT and AA dinucleotide steps in the minor groove. Together, our data define the biochemical requirements for MuvB interaction with promoters, reveal how this important cell cycle transcription factor complex engages DNA and demonstrate a novel structural mechanism for how transcription factors can access the DNA minor groove with sequence specificity. We propose a model in which DREAM binds cell cycle gene promoters at nucleosome-depleted transcription start sites and keeps genes repressed but poised for activation on cell cycle entry.

## Results

**LIN54's tandem CXC domains mediate DNA binding.** LIN54 contains two tandem CXC domains that are sufficient for binding a CHR site within the *CDK1* promoter[3]. To probe this interaction quantitatively and understand the significance of each domain, we performed fluorescence polarization (FP) analysis using purified LIN54 DBD and subdomain constructs. We titrated LIN54 DBD containing both CXC domains, DBD-N (the amino-terminal CXC domain only, residues 504–573) and DBD-C (the carboxy-terminal CXC domain only, residues 589–646) into a fluorescently labelled 27 bp DNA fragment from the *CDK1* promoter with a centrally located CHR motif[16] (called CHR27). Measuring changes in FP, we found that LIN54 DBD bound CHR27 with $K_d = 430 \pm 80$ nM (Fig. 1a). We observed that both DBD-N and DBD-C did not bind the duplex DNA under these conditions (Fig. 1a). DBD-N and DBD-C titrated simultaneously into CHR27 also did not bind the DNA (Fig. 1a). These results indicate that both CXC subdomains are necessary and must reside in the same polypeptide for high-affinity binding to DNA.

We considered that DBD-N or DBD-C might individually have weak DNA-binding affinity that is not observable by FP analysis. To test this possibility, we generated [15]N-labelled DBD-N or DBD-C, and monitored binding by nuclear magnetic resonance (NMR) to a 13 bp DNA fragment from the *CDK1* promoter (CHR13), which binds the entire DBD with slightly weaker affinity than CHR27 by FP (Fig. 1b,c and Supplementary Fig. 1a). Both individual subdomains yielded heteronuclear single quantum coherence (HSQC) spectra with dispersed peaks that are characteristic of well-structured proteins and we acquired sequence-specific backbone chemical shift assignments for DBD-C. On CHR13 titration, peak positions changed in both spectra in a manner consistent with fast or intermediate exchange kinetics and weak binding (Fig. 1b,c and Supplementary Fig. 1b,c). From the change in chemical shift values of 14 peaks in DBD-C, we calculated an average $K_d$ of $500 \pm 100$ μM (Supplementary Fig. 1d). The changes in peak positions in fast exchange in the DBD-N titrations were too small (< 0.05 p.p.m.) to determine affinity (Fig. 1b and Supplementary Fig. 1b). Taken together, our binding data indicate that individual DBD subdomains are capable of weak interactions with the CHR site, and that their tethering within the same polypeptide chain increases affinity.

**DNA sequence requirements for LIN54 binding.** We next identified the precise DNA sequence elements required for high-affinity LIN54 binding. We used an isothermal titration calorimetry (ITC) assay to measure affinity of LIN54 DBD for CHR13 and CHR13 variants with specific base pair changes in and around the central 5′-TTTGAA-3′ sequence (Fig. 2). LIN54 DBD binds CHR13 with $K_d = 2.8 \pm 0.1$ μM (Fig. 2a). This affinity is similar to that measured for CHR13 by FP ($K_d = 5 \pm 2$ μM; Supplementary Fig. 1a) but weaker than the affinity for CHR27. These results indicate that LIN54 has higher affinity for longer stretches of duplex DNA, while the majority of stabilizing interactions are within the short sequence containing the 5′-TTTGAA-3′ element. We continued analysing the shorter CHR13 sequence, as it was more suitable for structural studies.

We used single and dinucleotide variants to examine the important features of the 5′-TTTGAA-3′ sequence required for LIN54 affinity in the ITC assay (Fig. 2b). The *CDK1* promoter sequence used for CHR13 has two potential LIN54 DNA-binding frames, shown as solid (primary) and dashed (alternative) boxes in Fig. 2b. We examined the effects of mutating the first two thymines (sequences 1–4 in Fig. 2b) and the final two adenines (sequences 5–8) of the primary DNA-binding frame and found

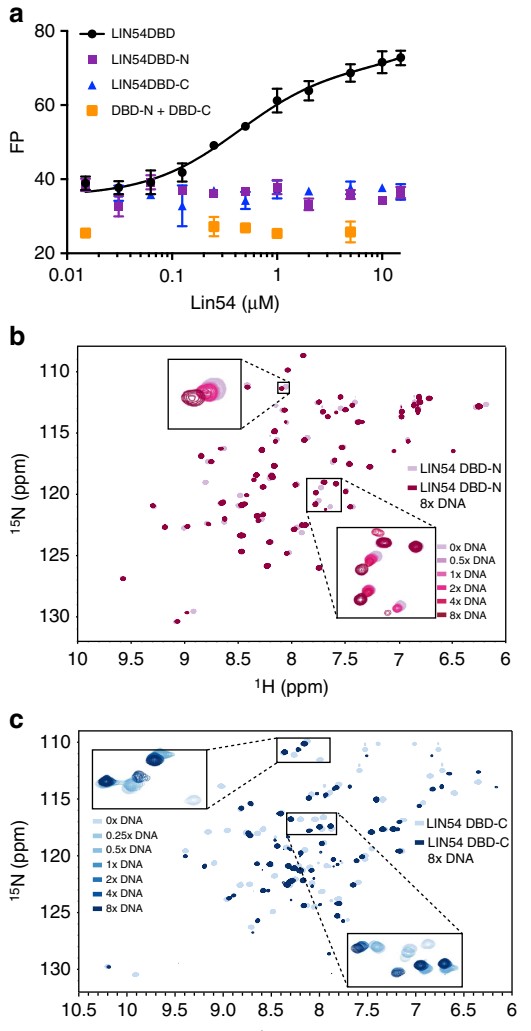

**Figure 1 | Both CXC domains of LIN54 are required for high-affinity DNA binding.** (**a**) FP analysis of LIN54 DBD binding to TAMRA-labelled CHR27 DNA from the *CDK1* promoter. The complete DBD binds CHR27 with affinity $K_d = 430 \pm 80$ nM, whereas the individual CXC domains (DBD-N and DBD-C) do not show binding in this assay or when titrated together. Error bars show the s.d. for three experimental replicates. (**b,c**) HSQC NMR spectra of 15 N-labelled 100 μM DBD-N (**b**) and 440 μM DBD-C (**c**). Each spectrum shows peak dispersion that is typical of a folded domain and the peaks change on titration of CHR13 DNA in a manner that is consistent with weak affinity binding. For clarity, only the full spectra in the absence and presence of 8 × DNA concentration are shown. The complete spectra across DNA concentrations are shown in Supplementary Fig. 1.

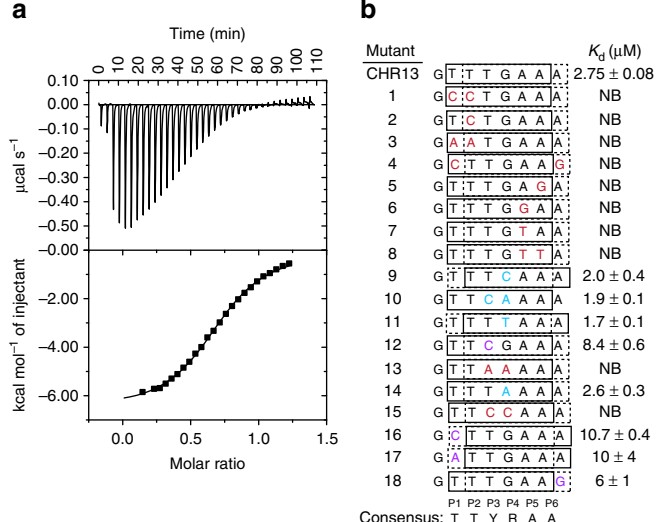

**Figure 2 | Biochemical identification of a CHR consensus sequence.** (**a**) ITC data of LIN54 DBD binding to CHR13. (**b**) ITC affinities of the LIN54 DBD for CHR13 and sequence variants, which are numbered for discussion in the main text. Each CHR13 sequence contains two potential binding sequences that are frame shifted by a single nucleotide. Both the putative best binding sequence (solid box) and potential alternate binding sequence (dashed box) are indicated. Mutations that do not change affinity are highlighted in blue, whereas mutations that reduce affinity are highlighted in purple and mutations that eliminate binding are highlighted in red. NB, no detectable binding.

specificity arises from the central six nucleotide positions. From these experiments, we were able to define a LIN54 DNA-binding motif of TTYRAA, where Y is a pyrimidine base and R is a purine base. We also note that there is an additional preference for at least one of the central positions (P3 or P4) to have an A–T base pair (for example, sequence 12 has lower affinity than sequences 10 and 14). These results align well with studies that identify CHR sequences in DREAM and Myb–MuvB-binding sites throughout the genome[13,16].

**Crystal structure of the DBD–CHR13 complex.** To understand how LIN54 recognizes CHR promoters and the molecular basis of our observed sequence specificity, we determined the crystal structure of LIN54 DBD bound to CHR13 at 2.4 Å resolution (Table 1 and Figs 3 and 4). The asymmetric unit contains two LIN54–DNA complexes that have nearly identical structures (C-α root-mean-squared deviation (RMSD) = 0.33 Å; Supplementary Fig. 2). Consistent with our NMR data, the DBD contains two independently folded subdomains corresponding to DBD-N and DBD-C (Fig. 3a). Both subdomains contain a CXC fold, which we define based on structural similarity to the CXC domain from MSL2 (Supplementary Fig. 3b,c), followed by an α-helix that contributes to the subdomain structural core (Fig. 3b,c). The subdomains both bind to the minor groove of CHR13, both on one side of the DNA helix. Electron density corresponding to the 15 amino acids linking the subdomains is lacking and there are no observable protein–protein contacts between them, which suggests that the subdomains are connected through a flexible tether.

The CXC folds present in the LIN54 DBD align well with each other (C-α RMSD = 0.35 Å) and with the CXC domain of MSL2 (C-α RMSD of 0.45 and 0.47 Å for DBD-N and DBD-C, respectively) (Supplementary Fig. 3a,b). The presence of Zn in LIN54 was confirmed by a fluorescence X-ray scan of the crystal at the synchrotron (Supplementary Fig. 4). Within the CXC fold,

that they are strictly required for binding. For example, a thymine to cytosine change in either of the first two positions (sequences 1, 2 and 4) or adenine to guanine change in either of the last two positions (sequences 5 and 6) results in complete loss of heat in the ITC titration. In the middle two positions, although mutation of the thymine in the third position (P3) to a cytosine does not affect binding affinity (sequences 9 and 10), mutation to an adenine or shifting the binding frame to place guanine in that position (sequences 13, 16 and 17) results in loss of some binding affinity. Conversely, mutation of the guanine in P4 to an adenine (sequence 14) preserves affinity, whereas mutation to cytosine (sequence 15) results in loss of detectable binding. Although we did observe some relatively modest effects of changing bases outside the CHR sequence (sequences 16–18), most of the

**Table 1 | Data collection and refinement statistics.**

| | |
|---|---|
| *Data collection* | |
| Space group | P1 |
| Cell dimensions | |
| $a$, $b$, $c$ (Å) | 39.3, 53.9, 67.1 |
| $\alpha$, $\beta$, $\gamma$ (°) | 98.4, 102.5, 106.6 |
| Resolution (last shell), Å | 36.03–2.42 (2.507–2.42) |
| $R_{merge}$* (last shell), % | 6.0 (40.5) |
| $I/\sigma$ (last shell) | 8.9 (2.1) |
| Multiplicity | 2.6 |
| $CC_{1/2}$ | 0.997 (0.732) |
| Completeness (last shell), % | 94.5 (95.2) |
| | |
| *Refinement* | |
| Resolution, Å | 36–2.42 |
| Reflection measured/unique | 46920/18229 |
| $R_{work}/R_{free}$† , % | 18.1/23.8 |
| No. atoms | 2701 |
| No. atoms: protein/DNA | 2633 |
| No. atoms: water | 56 |
| Average B-factor | 61.5 |
| Root mean squared deviation bond lengths, Å | 0.01 |
| Root mean squared deviation bond angles, ° | 1.21 |
| Ramachandran statistics (% most favoured/allowed/additionally allowed/disallowed) | 94/0/6/0 |

One crystal was used for the structure. Highest-resolution shell is shown in parenthesis.
*$R_{merge} = \Sigma\Sigma j|Ij - <I>|\Sigma Ij$, where $Ij$ is the intensity measurement for reflection $j$ and $<I>$ is the mean intensity for multiply recorded reflections.
†$R_{work}/R_{free} = \Sigma||F_{obs}| - |F_{calc}||/|F_{obs}|$, where the working and free R factors are calculated by using the working and free reflection sets, respectively. The free R reflections (5% of the total) were held aside throughout refinement.

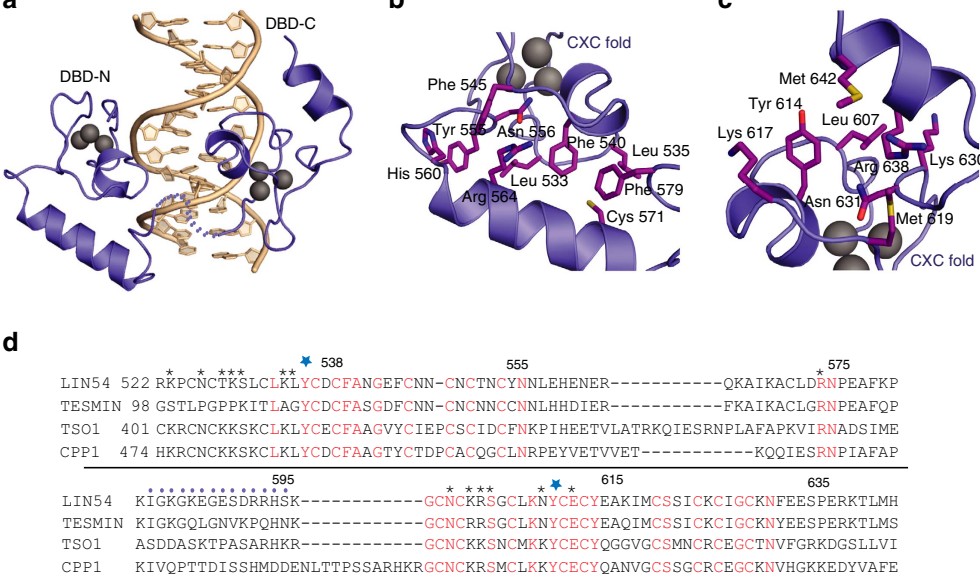

**Figure 3 | Structure of the LIN54 DBD.** (**a**) Overall structure of the LIN54 DBD in complex with CHR13. Zinc atoms are shown as grey spheres. The flexible linker between the domains is shown as dots. (**b,c**) Close-up views of DBD-N (**b**) and DBD-C (**c**) show the interactions between the CXC fold and the C-terminal helix, which form the structural core of each subdomain. (**d**) Sequence alignment of human LIN54, human tesmin, *Arabidopsis* TSO1 and soybean CPP1 domains shows conserved residues (red), residues that contact the CHR DNA (asterisks) and the critical tyrosines that bind the minor groove (blue stars). Residues in the flexible linker are indicated by purple dots.

three Zn ions are coordinated by nine cysteines, which are present in loops and one short α-helix. The polypeptide wraps around the three ions in one and a half turns of a right-handed helix. Beyond the cysteine coordination, the fold is stabilized by backbone hydrogen bonds, resembling interactions in a β-helix structure and several key sidechain contacts. For example, Asn556 and Asn631, which are conserved in MSL2, make hydrogen bonds

with the backbone carbonyl oxygens of Glu544 and Ile618, respectively.

Each DBD subdomain contains a C-terminal α-helix, not present in MSL2, which covers hydrophobic sidechains radiating from the CXC fold (Fig. 3b,c). In DBD-N, for example, Leu533, Leu535, Phe540, Phe545, Tyr555 and Asn556 from the CXC fold pack against one face of the helix (Fig. 3b). In DBD-C, the

interface is less extensive and is formed by Leu607, Tyr614, Met619, Lys630 and Asn631 from the CXC fold (Fig. 3c). In both subdomains, the interface-forming sidechains from the CXC fold are generally conserved in the tesmin family but not in MSL2 (Fig. 3d and Supplementary Fig. 3c). In contrast, residues in the C-terminal helix that form the interface are less conserved even among tesmin proteins, although the sequences all show propensities for helix formation by secondary structure prediction.

**LIN54 DNA-binding interactions.** The DBD subdomains bind adjacent sites in the minor groove with pseudo twofold symmetry. Almost all the protein–DNA contacts are within the central eight DNA base pairs, including the 5′-TTTGAA-3′ CHR motif. DBD-C binds the 5′-TTT half of the motif, whereas DBD-N binds the 3′-GAA half. Unambiguous electron density corresponding to the central bases in the asymmetric 5′-TTGAA-3′ sequence confirms that this orientation is uniform throughout the crystal structure (Supplementary Fig. 5). The CHR13 DNA in the crystal structure forms a standard B-form helix with little

perturbation to the DNA backbone. The protein–DNA complex is stabilized by an extensive set of backbone and sidechain hydrogen bonds to the DNA phosphate backbone and by insertion of a tyrosine from each subdomain into the minor groove. The DNA-binding interactions are mediated by residues conserved among the tesmin domain containing proteins (Figs 4b and 3d).

Each subdomain in LIN54 binds both DNA strands, using a distinct structural element for each strand (Fig. 4a,b). An N-terminal loop in the CXC fold makes several hydrogen bonds in which a backbone amide or sidechain hydrogen is donated to the phosphate backbone. In DBD-N, backbone amide hydrogens from Asn526, Thr528, and Lys529 and sidechains from Asn526 and Ser530 contact the backbone of A8' and A9'. In DBD-C, Asn600, Lys602 and Arg603 donate backbone hydrogens and Asn600 and Ser604 donate sidechain hydrogens to G7 and A8. A K(L/N)Y motif in each subdomain, which includes the inserted Tyr, binds the strand complementary to the strand contacted by the N-terminal loop. In DBD-N, Lys534 hydrogen bonds with the C11 phosphate and Leu535 is in van der Waals contact with the ribose of A9 and phosphate of A10. Arg574, which is in the

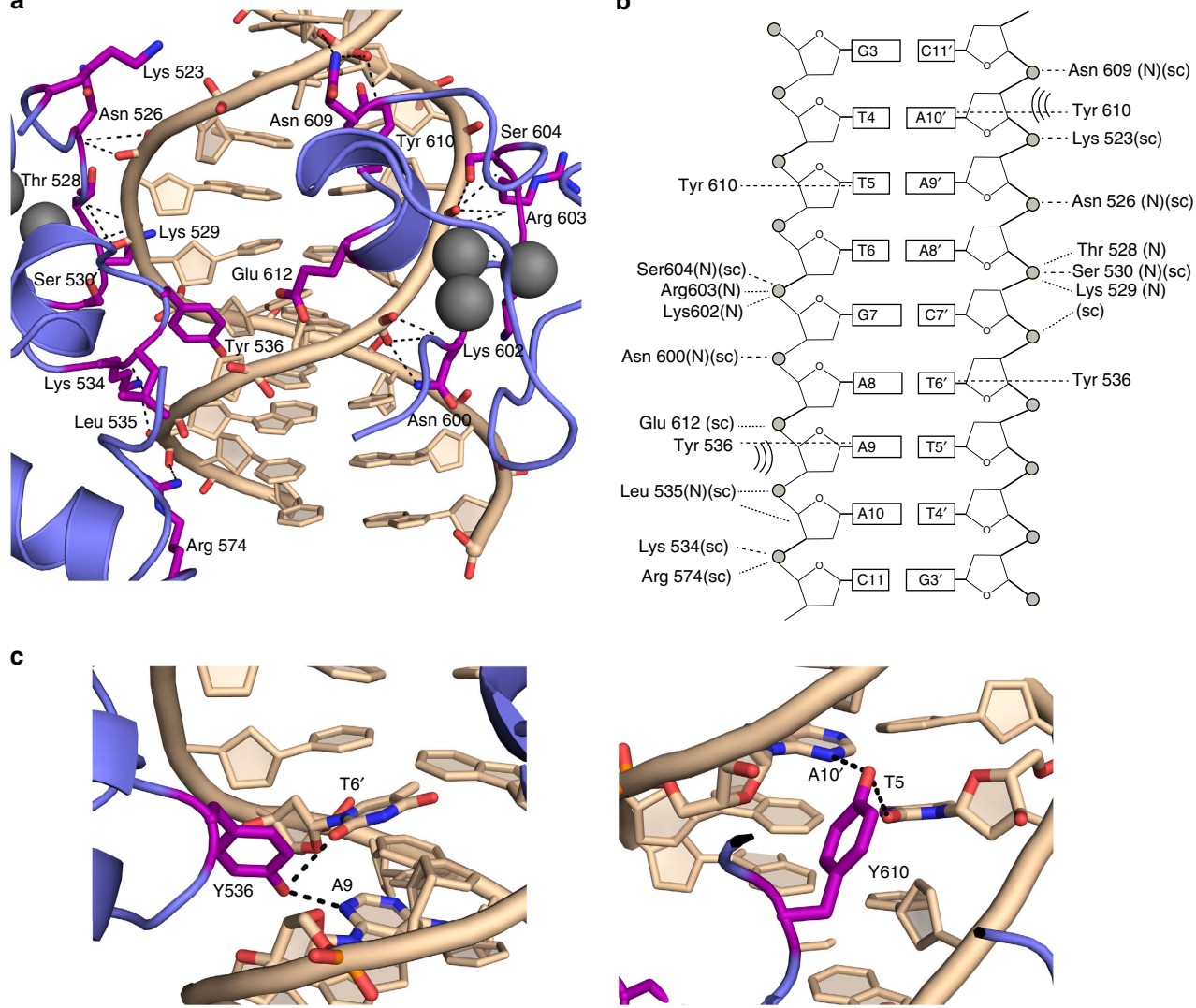

**Figure 4 | LIN54–DNA interactions reveal the structural basis for CHR recognition.** (**a**,**b**) DBD-N and DBD-C each bind one half of the CHR consensus, with residues from each subdomain primarily making hydrogen bond (dashed lines in schematic) and van der Waals (dotted lines) contacts to the DNA backbone. (**c**) Y536 and Y610 are both inserted into the minor groove and make specific hydrogen bonds to a TT/AA dinucleotide step.

C-terminal helix of DBD-N, hydrogen bonds with the C11 phosphate. In DBD-C, Asn609 makes both sidechain and backbone hydrogen bonds with C11'.

**Tyrosine–DNA interactions confer LIN54-binding specificity.** The most striking feature of the LIN54–DNA complex is the interaction of Tyr536 and Tyr610 with the minor groove, where those tyrosines bind both halves of the CHR consensus symmetrically (Fig. 4). Each tyrosine makes hydrogen bonds with two bases, one base from each strand and one step apart. Tyr610 binds the TT step at the beginning of the CHR consensus, whereas Tyr536 binds the AA step at the end. Thus, there are four total bases in the minor groove contacted, with two base pairs of bridging DNA (Fig. 4b,c). Additional van der Waals interactions between the tyrosine rings and deoxyribose sugars from both sides of the minor groove are observed, and the CH4' of sugars on opposing strands appear positioned for a CH–π interaction (Supplementary Fig. 6)[25,26]. The average minor groove width at

the points of tyrosine insertion is $9.9 \pm 0.5$ Å (measured phosphate to phosphate), which is narrow but typical of A/T tracts[27,28]. The narrow space optimizes the interactions between the sugars and tyrosine (Supplementary Fig. 6). Binding of the tyrosines also perturbs base pair structural parameters relative to those predicted for a standard B-form helix with the same sequence (Supplementary Fig. 7a)[29,30]. Specifically, the four T/A base pairs at the ends of the CHR consensus show large propeller twist angles and the four central base pairs show high buckle.

Tyrosine residues binding in the minor groove explains our observed LIN54 specificity for the consensus TTYRAA sequence (Fig. 4). The phenol group of Tyr610 makes hydrogen bonds with N3 of A10' and O2 of T5. A10' base pairs with T4, which is the most 5'-thymidine of the consensus (Fig. 4b,c). Tyr536 binds similarly to A9 and T6' (Fig. 4b,c). The position of the tyrosine required to satisfy both hydrogen bonds explains the need for a pyrimidine (Y) in P1 and P2, and purine (R) in P5 and P6. We propose that the strict requirement for T and A in these positions arises from their higher propensity for a narrow minor groove, which is critical for the van der Waals and CH–π interactions that stabilize the inserted tyrosines (Fig. 4b,c)[28,31]. T/A base pairs also have lower melting and stacking energies relative to C/G, which probably allows the propeller twist necessary for the tyrosine hydrogen binding (Supplementary Fig. 7a)[30,32]. In support of this hypothesis, we found that although LIN54 has no detectable affinity for sequences with C/G pairs in the first and last two positions of the CHR consensus, LIN54 binds weakly to sequences with C/inosine (I) pairs in those positions (Supplementary Fig. 7b). Similar to A/T pairs, C/I pairs only make two hydrogen bonds and are more susceptible to deformation. Similarly, we suggest that the requirement for Y in P3 and R in P4 arises in part because of the tolerance for buckling of a YR step, which also has relatively low stacking energy[29,30]. The preference for an A/T pair at either P3 or P4 is again likely to be due to the propensity for minor groove narrowing and tolerance to base step parameter deformation.

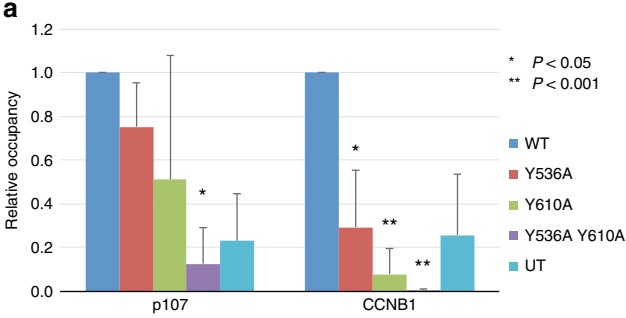

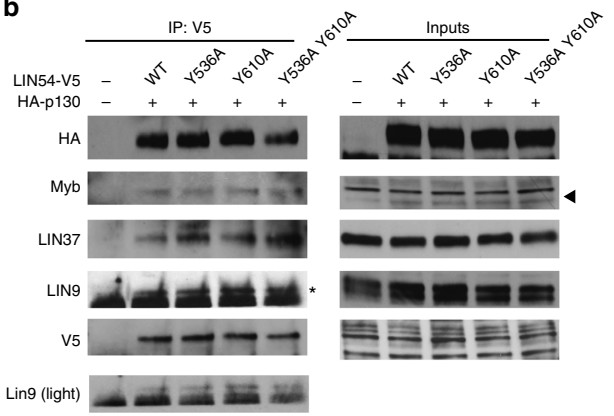

**Figure 5 | Critical DNA-binding tyrosines are necessary for MuvB recruitment to promoters. (a)** ChIP assay of V5-LIN54 and the indicated mutants after transfection into T98G cells. Following cross-linking and immunoprecipitation with an anti-V5 antibody, quantitative PCR was performed with primers specific to the *p107* and *CCNB1* promoters. The average promoter enrichment relative to wild-type LIN54 is reported and the error bars are s.d. from three biological replicates and one technical replicate. The *P*-value evaluating statistical significance was calculated for the promoter enrichment of a mutant relative to wild-type using a two-tailed Student's *t*-test. **(b)** Wild-type or mutant V5-LIN54 were co-transfected into T98G cells with HA-tagged p130, extracts were immunoprecipitated with an anti-V5 antibody and precipitates were probed for the indicated proteins. An IgG band in the LIN9 blot is marked with an asterisk. Triangle corresponds to BMYB band in the input samples. All bands shown in the input for Lin54-V5 are nonspecific as the protein was undetectable until after immunoprecipitation. The full blots are shown in Supplementary Fig. 9.

**LIN54 tyrosines are critical for MuvB promoter recruitment.** From the ITC assay we found that amino-acid substitution of either Tyr536 or Tyr610 with Ala or Phe resulted in loss of binding to CHR DNA (Supplementary Table 1). These results are consistent with the requirement for both domains to bind the DNA motif (Fig. 1a) and they demonstrate that the ability of tyrosine to form hydrogen bonds with specific DNA bases is critical. Most proteins that bind the minor groove of DNA use arginine[27] and we speculated that arginine might be able to substitute for tyrosine and form a similar hydrogen bond network with the DNA. However, substitution of arginine for either Tyr536 or Tyr610 resulted in a protein unable to bind DNA in the ITC experiment (Supplementary Table 1). Tyrosines at similar positions in the CXC domain are conserved in other homologous tesmin family members (Fig. 3d), suggesting that these proteins may bind DNA in a similar manner.

We next tested the effect of tyrosine mutations on recruitment of the MuvB complex to CHR promoters in cells. We performed chromatin immunoprecipitation (ChIP) to examine the association of transfected LIN54 with the *p107* and *CCNB1* promoters (Fig. 5a and Supplementary Fig. 8a), which were chosen to represent genes expressed early and late in the cell cycle, respectively. The *p107* promoter contains a CDE site (5'-TTGGCGC-3'), to which E2F can bind[17], and a non-consensus CHR site (5'-TTTGAG-3'), whereas the *CCNB1* promoter contains no CDE site and an optimal consensus CHR site (5'-TTTAAA-3'). The *CCNB1* promoter displayed significantly reduced occupancy of LIN54 containing individual

Y536A and Y610A mutations compared with wild type. Occupancy of the double mutant (Y536A Y610A) was significantly reduced at both the *p107* and *CCNB1* promoters. These observations are consistent with the impaired DNA binding of the mutant observed *in vitro* (Supplementary Table 1). We confirmed in T98G cells that the tyrosine

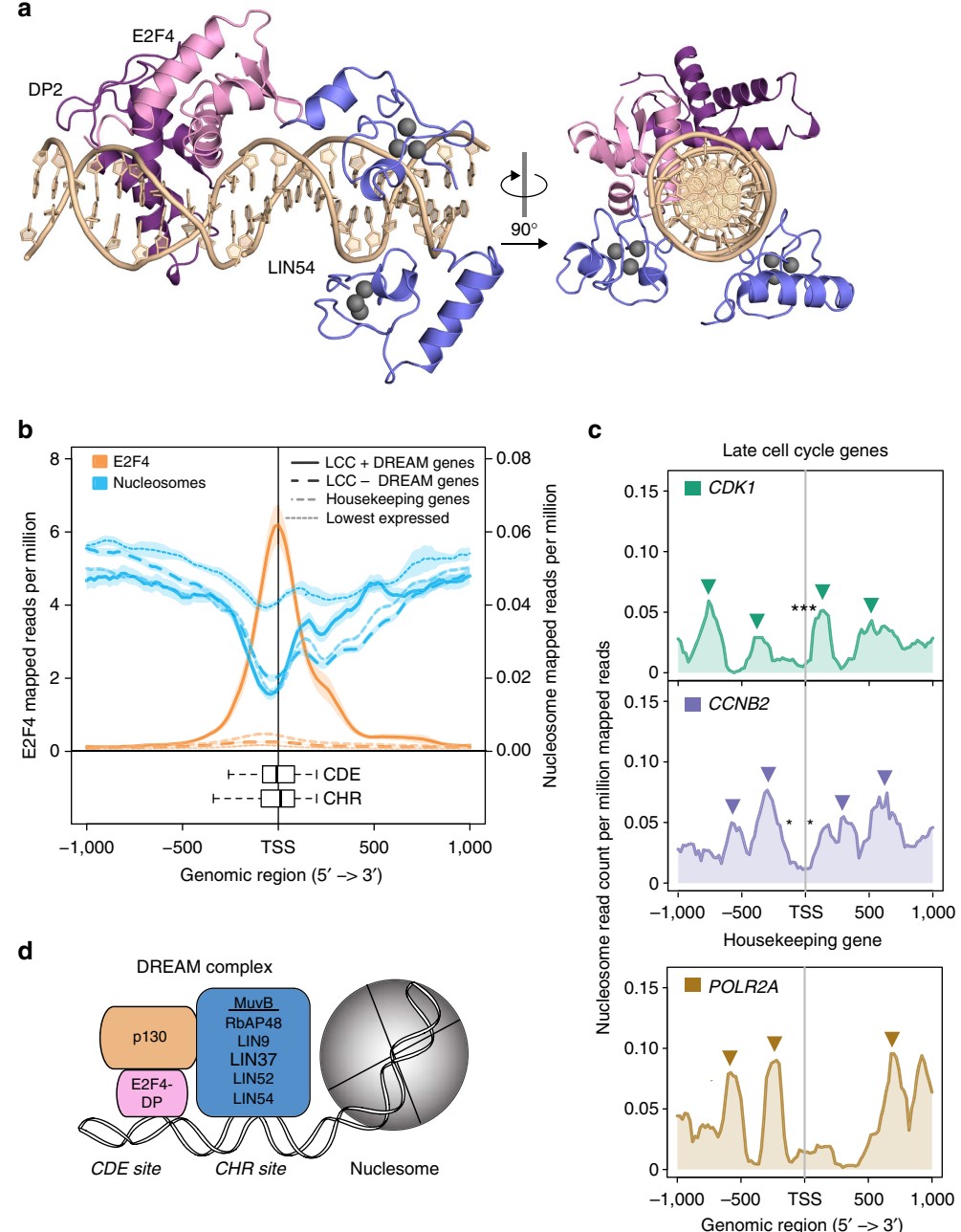

**Figure 6 | Nucleosomes are positioned downstream of DREAM promoter binding sites.** (**a**) Structural model of DREAM–DNA association constructed from the LIN54 DBD-CHR13 crystal structure from our study and the previously determined structure of the E2F4-DP2 DBD heterodimer bound to a CDE-like DNA sequence (1CF7.pdb). The LIN54 DBD binds the minor groove, while E2F4-DP2 DBDs bind the major groove. In this model built with four nucleotides separating the CHR and CDE elements, the DBDs bind to opposite sides of the helix. (**b**) Average read density profiles of E2F4 (orange) and nucleosomes (blue) 1,000 bp upstream and downstream of the TSS of 155 late cell cycle genes bound by the DREAM complex and containing a TTYRAA CHR motif (bold line) and 900 late cell cycle genes not bound by the DREAM complex and not containing a TTYRAA CHR motif (dashed line). The read density profile of nucleosomes and E2F4 surrounding 3,804 housekeeping genes (light dashed line) and 1,732 genes with low expression (dotted line) are included as references. The semi-transparent shade around the mean represents the standard error of the mean across the regions. The positions of the CHR (TTYRAA) and CDE (SGCGCGS) near the TSS of DREAM-bound late cell cycle genes are shown. Box plots indicate the median position (black bar), the 25th–75th percentile (box) and the 2.5th and 97.5th percentile (extended whiskers). (**c**) Read density profiles of nucleosomes surrounding the TSS of two late cell cycle genes bound by DREAM and containing a TTYRAA CHR: *CDK1* (encodes CDK1) and *CCNB2* (encodes Cyclin B2). A housekeeping gene, *POLR2A* (encodes RNA Pol II), is shown for comparison. Triangles indicate approximate locations of well-positioned nucleosomes. Asterisks indicate approximate locations of a CHR motif. (**d**) Schematic representation of the DREAM complex assembled on a promoter containing both CDE and CHR sites with a nucleosome nearby in the body of the gene.

mutations do not affect the assembly of LIN54 into the Myb–MuvB and DREAM complexes. Similar to wild-type LIN54, V5-tagged LIN54 tyrosine mutants co-immunoprecipitated other MuvB components (LIN37 and LIN9), Myb and co-transfected p130 (Fig. 5b and Supplementary Figs 8b and 9). The observation that the double mutant influences the LIN54 occupancy at the *p107* promoter despite the presence of a poor CHR site suggests that weak LIN54–DNA interactions may stabilize DREAM binding to the promoters that also contain CDE E2F-binding sites. Notably, in HeLa cells we observe a significant effect of the mutation on LIN54 occupancy at the *CCNB1* promoter but not at the *p107* promoter (Supplementary Fig. 8). The DREAM complex does not assemble in HeLa cells due to the presence of HPV E7 protein[6,33,34]. This result is consistent with the notion that LIN54 binding to the *p107* promoter is primarily driven by E2F binding to the CDE in the context of DREAM, with an additional contribution from a weak interaction between LIN54 and the suboptimal CHR site.

**DREAM-bound promoters have nucleosomes downstream.** DREAM frequently localizes to promoters containing both a CHR and a CDE[17,18]; the CDE serves as an E2F-DP-binding site[17,35]. Although a CDE is not a requirement for DREAM binding, many promoters repressed by DREAM contain a CDE four base pairs upstream of the CHR[18]. We used the novel LIN54 DBD-CHR structure together with the previously determined structure of the E2F4-DP2 DBD bound to an E2F promoter, to create a structural model for DREAM bound to DNA (Fig. 6a). Consistent with reported footprinting data[36], LIN54 contacts the minor groove and E2F-DP binds the major groove. The model shows that LIN54 and E2F-DP bind the DNA helix on opposite faces. We found that three or four base pairs is the closest the two DNA sequence elements can be without steric clashes between the two bound DBDs.

The organization of the DBDs on opposite faces of the DNA helix suggests that DREAM and histones cannot simultaneously bind DNA, and that DREAM probably associates with nucleosome-free DNA. We examined raw data from the ENCODE Consortium[37] to explore this possibility. Specifically, E2F4 ChIP sequencing (ChIP-seq) and microccocal nuclease (MNase)-seq nucleosome positioning data from a human cell line were mapped to the human genome. We compared the average position of nucleosomes near the transcriptional start site (TSS) of known late cell cycle genes that contain a TTYRAA CHR and are bound by DREAM, late cell cycle genes that lack a CHR and are not bound by DREAM, housekeeping genes and a set of genes with low expression (Fig. 6b and Supplementary Fig. 10)[16,38,39]. As predicted by the structural model, the positions of E2F4 and the CHR sites at the TSS overlap with a region of MNase sensitivity that is indicative of the absence of protective histones. The MNase sensitivity at the TSS of DREAM-bound genes is greater than at the TSS of genes with low expression, which is a notable difference considering that DREAM is known to repress transcription. The data also show that late cell cycle genes bound by DREAM have higher nucleosome occupancy downstream of the TSS than do late cell cycle genes not bound by DREAM and housekeeping genes. For example, both *CDK1* (encodes cyclin-dependent kinase 1) and *CCNB2* (encodes Cyclin B2), known DREAM-regulated late cell cycle genes, have a nucleosome positioned downstream of and near the TSS (Fig. 6c). In contrast, the housekeeping gene *POLR2A* (encodes RNA polymerase II) lacks a nucleosome positioned near the TSS (Fig. 6c). We conclude that DREAM-bound promoters have nucleosome profiles that resemble expressed genes around the TSS (nucleosome free) but resemble repressed genes (nucleosome bound) in the downstream coding region.

## Discussion

The CHR element is present in the promoter of many genes that are expressed in a cell cycle-dependent manner and is required for proper regulation of transcription. From genomic approaches, the consensus sequence for CHR sites is thought to be 5′-TTTGAA-3′. However, numerous promoters with non-canonical CHR sites have been identified as regulated by LIN54 binding[16]. From the LIN54 DNA-binding measurements described here, the CHR motif that results in optimal LIN54 binding is TTYRAA, where either the pyrimidine (Y) is a T or the purine (R) is an A[16]. Notably, a change in one of the outside two bases (TT or AA) results in dramatically decreased overall affinity. Approximately 79% of DREAM-bound late cell cycle promoters[16] contain the defined optimal CHR sequence. Our ChIP data suggest that DREAM binding to the *p107* promoter, which contains a non-canonical CHR site, is still influenced by LIN54–DNA interactions. The *p107* promoter (5′-TTTGAG-3′) and other promoters such as MELK (5′-TTTGAT-3′), RAD18 (5′-TTCGAG-3′), and RAD54L (5′-TTCGAT-3′), which bind DREAM but lack a canonical CHR[16], have at least half of the consensus sequence intact and contain a proximal CDE site. We suggest that DREAM binding at these promoters can be influenced by both E2F-CDE binding and additional, weak interactions between LIN54 and the noncanonical CHR.

Multivalent interactions with DNA promoters are likely to be a critical mechanism for enabling a small set of cell cycle transcription factors to combinatorially regulate a large number of genes with specificity and precise timing of activation, yet the biochemical details of this process are unclear. Our data support the hypothesis that DREAM uses multiple DBDs, to engage dual promoter sites with high-affinity bipartite association, with the local concentration of individual DBDs being increased by complex formation through the DREAM core (Fig. 6d). Our structural model for a dual-site promoter positions the C-termini of LIN54, E2F4 and DP all within close proximity on one side of the DNA helix (Fig. 6a). This organization is consistent with how the protein DBDs are predicted to connect back to the rest of the DREAM complex (Fig. 6d)[2,3,6,40].

We used our structural model and available ChIP-seq and MNase-seq data to begin to investigate the architecture of DREAM promoters and how expression at these promoters is regulated. We found that DREAM binding coincides with nucleosome-depleted regions at transcription start sites that are just upstream of regions with high nucleosome occupancy (Fig. 6b,c). We speculate that the role of DREAM may be to hold repressive nucleosomes in place, perhaps through the RBAP48 histone-binding domain in MuvB. Promoters bound by MuvB may be kept in a poised state in which genes are repressed until the appropriate time in the cell cycle and then activated by dissociation of DREAM, formation of Myb–MuvB and remodelling of downstream nucleosomes. Our structural model also supports previous suggestions that additional binding factors such as E2F-DP and Myb could function to refine the localization of MuvB to a specific set of target genes depending on the cell cycle stage[16–18,36]. The activating Myb–MuvB complex binds CHR elements in S, G2 and M, and independently of a CDE; some of these promoters also contain a Myb-binding site[16,18].

The organization of tandem CXC-containing domains that comprise the LIN54 DBD is found in other tesmin family proteins[19,41], including TSO1 from *Arabidopsis*, and CPP1 from soybean. The tyrosines that recognize the CHR sequence and most of the residues that contact the DNA backbone are conserved (Fig. 3d); thus, it is likely to be that these proteins all share a similar DNA binding and recognition strategy to LIN54. In contrast, MSL2 contains a single CXC domain, instead of the

tandem CXC domains found in tesmin proteins. In addition, MSL2 does not contain the C-terminal helix that completes each LIN54 subdomain and it binds the minor groove through insertion of an arginine.

The two LIN54 subdomains DBD-N and DBD-C do not share a protein–protein interaction interface, yet both are required for high-affinity DNA binding. It is likely to be that the distance constraint imposed by the length of the flexible connector between the two CXCs and the local concentration of CXC domains creates an environment favourable to DNA sequence recognition and high-affinity binding. Although it is clear in our crystal structure that DBD-C and DBD-N bind the 5′ and 3′ halves of the CHR consensus, respectively, the origin of this specificity is not obvious from the observed DNA–protein contacts. It is not certain from our binding data whether this specific orientation always exists in solution. Our NMR data indicate that DBD-C binds DNA with higher affinity than DBD-N. We also note that the 5′-half of the CHR consensus, to which the DBD-C in the crystal structure binds, is present in many non-canonical sites within promoters that bind DREAM, including *p107* (ref. 16). These observations suggest that DBD-C may consistently bind the 5′-TTY sequence and further structural studies of the entire DREAM complex may inform whether this directionality is important for assembly and function.

Our structural and biochemical data indicate that the LIN54 DBD specifically recognizes the CHR motif using two tyrosine residues. The recognition of nucleotide sequences by tyrosine insertion into the DNA minor groove is rare among DNA-binding proteins, although our search of the protein database identified a few other examples in transcription factors[27,42–44]. Most sequence-specific DBDs use secondary structural elements to recognize DNA sequences in the major groove[27]. One common feature of DNA minor groove-interacting proteins is the use of arginine to indirectly read DNA sequences by recognizing the narrow minor groove sequences enriched for A and T[27,29,30,32]. The discovery of LIN54's interesting tyrosine-based DNA-binding mode expands our knowledge of minor groove-binding proteins and may implicate tesmin family proteins in previously uncharacterized cellular functions.

## Methods

**Protein expression.** The human LIN54 DBD (S515-D646), DBD-N (S515-D574) and DBD-C (G589-D646) were expressed in and purified from *Escherichia coli* BL21 cells as N-terminal glutathione *S*-transferase fusion proteins with TEV cleavage sites. In mid-log phase, 200 μM ZnSO₄ was added to the cells followed by induction with 1 mM isopropylthiogalactoside. Cells were grown overnight at 20 °C. Fusion proteins were purified from lysates with glutathione sepharose affinity chromatography. The elution fraction was cleaved with TEV protease and buffer exchanged using dialysis into 10 mM Tris, 1 M NaCl and 5 mM dithiothreitol pH 8.0. The protein was then passed over glutathione sepharose resin to remove fee glutathione *S*-transferase, concentrated and run over Superdex-75 (GE Healthcare).

**Isothermal titration calorimetry.** Equilibrium dissociation constants and stoichiometry for LIN54 DBD binding to duplex DNA were obtained using ITC with the Micro Cal VP-ITC system. LIN54 DBD was run over Superdex-75 into 20 mM Tris, 150 mM NaCl and 0.1% Tween 20 pH 8.0. Oligonucleotides from IDT were dissolved in the same buffer at 1 mM concentration (CHR sequence 5′-GAGTTTGAAACTG-3′). Complementary oligonucleotides were annealed by mixing one to one then heating to 98 °C and slowly cooling to room temperature. Duplex DNA was titrated into LIN54 DBD at 50 μM at 25 °C. Reported $K_d$ values are the average fits from two or three biological replicates with the s.d. reported as error.

**FP-binding assay.** LIN54 DBD constructs were mixed with 10 nM of TAMRA-labelled duplex CHR DNA (5′-CCT TTA GCG CGG TGA GTT TGA AAC TGT AA/36-TAMSp/-3′) in a buffer containing 20 mM Tris, 150 mM NaCl and 0.1% Tween 20 pH 8.0. In a 384-well plate, 20 μl was used for each measurement. FP experiments were made in triplicate and measured using a Perkin-Elmer EnVision plate reader. Reported FP values were determined using the software from the

instrument as follows: $mP = 1,000 \times (S - G \times P)/(S + G \times P)$ where $S$ is the intensity of fluorescence parallel to excitation plane, $P$ is the perpendicular fluorescence intensity and $G$ is a correction factor to ensure positive ratio values. The data were plotted using a one-site total model in Prism Graphpad. Error bars represent s.d. between the three measurements. The reported error in $K_d$ is derived from curve fits.

**Crystallization and structure determination.** LIN54 DBD was prepared for crystallization by elution from a Superdex-75 (GE Healthcare) column in buffer containing 20 mM Hepes pH 7.0 and 200 mM NaCl. CHR13 duplex DNA (5′-GAGTTTGAAACTG-3′) was added in twofold molar excess to 11 mg ml⁻¹ of LIN54 DBD. The protein–DNA complex was crystalized either by sitting-drop or hanging drop vapour diffusion at 22 °C. Crystals formed over several days in 313 mM magnesium chloride hexahydrate and 21% PEG 3,350. Crystals were frozen in a solution of mother liquor and 25% ethylene glycol.

Data were collected at the Advanced Photon Source, Argonne National Laboratory at Beamline 23-IDB and at the Advanced Light Source, Lawrence Berkeley National Laboratory Beamline 8.3.1. Diffraction spots were integrated using MOSFILM[45] and data were merged and scaled using Scala[46]. Phases were solved by molecular replacement using Phaser[47]. A homology model for half of the LIN54 DBD was constructed using MSL2 (PDB: 4RKH) and deleting a small seven amino-acid loop region. First, generic B-form DNA was used as a search model, then the DNA solution was set as a fixed partial solution and one copy of MSL2 was used as a search model. That solution (DNA plus protein) was fixed as a partial solution and MSL2 was used as a search model for the additional half of the LIN54 DBD. The final model was built with Coot[48] and the models were refined with PHENIX[49]. Coordinates and the corresponding structure factors have been deposited in PDB under the accession code 5FD3.

**DNA shape analysis.** The CHR DNA in the crystal structure of the LIN54 complex was submitted to the w3dna server (http://w3dna.rutgers.edu/)[50]. To produce the ideal B-form DNA structure for comparison, the CHR13 sequence was submitted to the server, which calculated ideal B-form DNA structural parameters.

**Analysis of nucleosome positioning.** Raw reads from E2F4 ChIP-seq (GSE31477) and MNase-seq (GSE35586) experiments performed on the GM12878 lymphoblastoid cell line (ENCODE tier 1) were downloaded from the NCBI Gene Expression Omnibus[51]. These data were generated by the ENCODE Consortium[37]. Raw reads were mapped to the human genome (hg38) using BOWTIE[52]. Combined replicate data resulted in a total of 41,253,604 and 1,788,468,710 mapped reads for E2F4 and nucleosomes, respectively. Average E2F4 binding and nucleosome positioning profiles at specified promoter regions were generated using ngs.plot[53]. Late cell cycle (1,408), 745 DREAM-bound and 3,804 housekeeping gene promoter regions used in this analysis were defined previously[16,38]. GM12878 gene expression data (GSE26386) were used to define the 1,732 low expression gene promoter regions[39]. Motif locations were generated by searching for TTYRAA (CHR IUPAC code) and SGCGCS (CDE IUPAC code) within the specified promoter regions using the HOMER motif analysis tool[54]. This analysis identified two gene sets used in the promoter region profiles: 155 late cell cycle gene promoters bound by DREAM and containing TTYRAA and 900 late cell cycle gene promoters not bound by DREAM and not containing TTYRAA.

**ChIP and quantitative PCR analyses.** ChIP was performed using a modification of previously published method[55]. Approximately 10⁶ T98G or HeLa cells (from ATCC) were transiently transfected with pEF6 vectors encoding V5-tagged wild-type or mutant LIN54 alleles. After 24 h, the cells were treated with cross-linking buffer containing 11% formaldehyde, 0.1 M NaCl, 1 mM EDTA, 0.5 mM EGTA and 50 mM HEPES pH 8.0 for 10 min at room temperature. Cross-linking was stopped by adding 0.125 M glycine and the cells were scraped into PBS. Cells were collected by centrifugation and rocked at 4 °C for 10 min in a buffer containing 50 mM HEPES pH 7.5, 140 mM NaCl, 1 mM EDTA, 10% glycerol, 0.5% NP-40, 0.25% Triton X-100 and protease inhibitor cocktail. Insoluble pellets containing chromatin were collected by centrifugation, resuspended in buffer containing 200 mM NaCl, 1 mM EDTA, 0.5 mM EGTA, 10 mM Tris HCl pH 8.0 and protease inhibitors, incubated for 10 min at room temperature and then collected again by centrifugation at 1,000 g for 10 min. The resulting chromatin pellets were resuspended in sonication buffer (1 mM EDTA, 0.5 EGTA, 10 mM Tris HCl pH 8.0 and protease inhibitors) and sonicated on ice to obtain DNA fragments of an average length of 500 bp. Sonicated chromatin was extracted by adding 0.5% N-lauroyl-sarcosine for 10 min at room temperature and clarified by centrifugation at 14,000 g for 10 min at 4 °C. Chromatin (12.5 μg) was incubated overnight with 1 μg of anti-V5 antibody (AbD Serotec) at 4 °C and then mixed with protein A/G Dynabeads mixture (Novex) preblocked with sonicated salmon sperm DNA (Invitrogen). The beads were incubated with immune complexes for 3 h at 4 °C and then washed three times with RIPA buffer (0.5 M LiCl, 50 mM HEPES pH 7.5, 1 mM EDTA, 1% NP-40, 0.7% sodium deoxycholate and protease inhibitors) and three times with EBC buffer (50 mM Tris HCl pH 7.4, 150 mM NaCl and 0.5% NP-40). Immune complexes were then eluted by incubating the beads in 100 μl of TES elution buffer (50 mM Tris at pH 8, 10 mM EDTA and 1% SDS) at 65 °C for 20 min

followed by centrifugation. Cross-links were reversed by overnight incubation at 65 °C, followed by treatment with 20 µg RNAse A (Qiagen) at 37 °C for 1 h and 40 µg Proteinase K (Roche) at 65 °C for 2 h.

Resulting DNA samples were purified using QIAquick spin columns (Qiagen) and analysed by quantitative PCR in triplicate using SYBR Green dye (Applied Biosciences) and the following human promoter-specific primer pairs: p107 (5′-AGGCAGACGGTGGATGACAACAC-3′, 5′-TCAGCGTGGGGCTTGTCCTCGAA-3′) and CCNB1 (5′-CGATCGCCCTGGAAACGCATTC-3′, 5′-CCAGCAGAAACCAACAGCCGTTC-3′). For each condition, fold enrichment was calculated as % of input chromatin and the relative enrichment was determined using $\Delta\Delta C_t$ method as described previously[5].

**Protein co-immunoprecipitation.** T98G cells were transiently transfected with pEF6 vectors encoding V5-tagged wild-type or mutant LIN54 alleles and pcDNA3.1 vectors encoding haemagglutinin (HA)-tagged wild-type p130. Cells were extracted using EBC lysis buffer (50 mM Tris HCl pH 7.4, 150 mM NaCl, 0.5% NP-40, protease and phosphatase inhibitor cocktails), immunoprecipitated using anti-V5 antibody (AbD Serotec) and subjected to western blot analysis using rabbit antibodies specific for human LIN9 and LIN37, made in collaboration with Bethyl, anti-Myb (sc-724) (Santa-Cruz Biotech), rabbit anti-V5 (Bethyl Inc.) or mouse monoclonal anti-HA (12CA5), rabbit anti-HA (sc-805) (Santa-Cruz Biotech)[2]. The LIN9, LIN37, Myb and V5 antibodies were used at 1:1,000 dilution. The HA antibody was used at 1:15,000 dilution. Full western blottings are shown in Supplementary Fig. 9.

**NMR experiments.** NMR experiments were conducted at 25 °C on a Varian INOVA 600-MHz spectrometer equipped with $^1$H, $^{13}$C, $^{15}$N triple-resonance, $z$ axis pulsed-field gradient probes. All samples were prepared in a buffer containing 35 mM MES, 75 mM NaCl and 5% D2O (pH 6.0). All NMR data were processed with NMRPipe and NMRDraw[56]. Chemical-shift assignments were made with SPARKY (https://www.cgl.ucsf.edu/home/sparky/) with NMR data obtained from standard three-dimensional triple-resonance experiments acquired on 580 µM uniformly $^{13}$C, $^{15}$N-labelled DBD-C protein, including HNCO, HNCACB and CBCA(CO)NH spectra. $^{15}$N-HSQC titration of 100 µM $^{15}$N-DBD-N or 440 µM $^{15}$N-DBD-C proteins was done by stepwise addition of CHR13 DNA from a stock that was 20 mM in water. Samples were concentrated to 600 µl final volume and adjusted to a final concentration of 5% (v/v) D2O. $^{15}$N HSQC titration data were analysed with NMRViewJ[57], with chemical-shift perturbations defined by the equation $\Delta\delta_{tot} = [(\Delta\delta_{1H})^2 + (\chi(\Delta\delta_{15N})^2]^{1/2}$ and normalized with the scaling factor $\chi = 0.17$, established from estimates of atom-specific chemical-shift ranges in a protein environment[58].

**Data availability.** Protein structure data are available from RCSB Protein Databank: Lin54–5FD3 (this paper), MSL2–4RKH[24] and E2F-DP—1CF7 (ref. 35). NMR backbone assignments for LIN54 DBD-C are available from BMRB accession number 26810. MNase-seq data for nucleosome positioning are available from NCBI Gene Expression Omnibus—GSE35586 (ref. 37) and E2F4 ChIP-seq data are available from NCBI Gene Expression Omnibus—GES31477 (ref. 37). All additional relevant data supporting the findings of this study are available from the authors on request.

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

## Acknowledgements

This research used resources of the Advanced Photon Source, beamline 23-ID-B, a U.S. Department of Energy (DOE) Office of Science User Facility operated for the DOE Office of Science by Argonne National Laboratory under Contract No. DE-AC02-06CH11357. Data collection at the ALS Beamline 8.3.1 is supported by the UC Office of the President, Multicampus Research Programs and Initiatives Grant MR-15-328599 and Program for Breakthrough Biomedical Research, which is partially funded by the Sandler Foundation. We thank Andrew Blair and Max Garcia for writing our in-house software to search the PDB. This work was supported by grants from the National Institutes of Health to L.L. (R01CA188571), S.S. (R01GM34059) and S.M.R. (R01CA132685).

## Author contributions

Conceptualization, A.M. and S.R. Methodology, A.M., L.L., S.S. and S.R. Formal analysis, P.G. Investigation, A.M., J.F. and A.I. Data curation, H.L. and S.T. Writing—original draft, A.M. and S.R.; review and editing, L.L and S.S. Funding acquisition, S.R., L.L. and S.S. Resources, S.R. and L.L. Supervision, S.R., L.L. and S.S.
