## [Peer Review File · Nature Communications]

Reviewer #1 (Remarks to the Author)

The manuscript by Marceau and coworkers presents structural analyses on LIN54 binding to CHR promoter elements. They produce recombinant fragments of the two CXC domains from human LIN54 in *E. coli* and assay them for binding to DNA representing various CHR elements. One result is that both subdomains are required for binding of LIN54 to CHR sites. Several variants of CHR sites were tested for their affinity to the dual CXC domain fragment by isothermal titration calorimetry ITC, which yields the preferred binding consensus TTYRAA. From the crystal structure it is observed that the two CXC subdomains fold independently and are tethered by a flexible link peptide. The authors find three Zn coordinated by nine cysteines. The contacts to DNA are made by residues conserved among proteins related to LIN54. Two tyrosines contact the DNA in the minor groove. In combining structures from E2F4/DP2 binding to E2F sites with the LIN54/CHR structure, the authors model the configuration of DREAM binding to CDE/CHR tandem elements. From the combined structure, the 4 base pair spacer between CDE and CHR elements can be explained. Furthermore, binding of DREAM to the two sites on opposite sides of the DNA suggests an exclusion of nucleosomes in this region, but DREAM appears to support nucleosome binding downstream of the transcriptional start site.

Comments:

Experimentally this manuscript is sound. The structural analyses were performed by x-ray crystallography and NMR analysis, yielding consistent data from both approaches. Mutant variants LIN54 fragments were tested by CHIP for binding. Binding results are consistent with the ITC data from the study.

The results from this study are important for the field. It is the first structural analysis of a LIN54 protein. Thus, it provides first insight into protein binding to CHR elements. These results place the importance of the study on the level of determining the structure of E2F complexes bound to E2F sites. Furthermore, if there were still any doubts that LIN54 is really the subunit of the MuvB complex contacting the DNA, the doubts will be resolved with the data presented.

It is important to mention that the results obtained by Marceau and coworkers are consistent with data obtained by different methods for the CHR site consensus and the configuration of CDE versus CHR elements.

It was stated in the manuscript that p130 and CCNB1 were chosen as examples because of their cell cycle-dependent expression. However, the particular CHR sites for the two genes are not presented. How do they correlate with the LIN54/CHR binding consensus? A CHR for CCNB1 has been published. Is there any reported CHR element for the p130 gene?

Minor comments:

The title of the manuscript is too general. It should be more specific and include the abbreviated names 'LIN54' and 'CHR'.

In the first paragraph of 'Discussion': '...we defined the CHR motif ... as TTYRAA...'. This should be worded differently as this consensus had been reported in Ref. 14, which has been referenced in several parts of the manuscript correctly.

Reviewer #2 (Remarks to the Author)

NCOMMS-16-00582

Structural Basis for Recognition of Cell Cycle Regulated Gene Promoters

Seth Rubin

This is an interesting manuscript that describes the structure of the LIN54 DNA binding domain bound to a CHR DNA motif. LIN54 is part of the MuvB complex involved in cell cycle gene regulation. The authors determined binding association for the LIN54-DBD and identified the CHR consensus sequence TTYRAA as the optimum sequence. They determined the crystal structure of and found that the two CXC folds were nearly identical to each other and bind to adjacent sites in the minor groove in a nearly symmetrical manner. Two tyrosine residues in the N- and C- terminal binding domain make important contacts with the DNA. Point substitutions of the LIN54-DBD tyrosine residues significantly decreases binding to the p107 (RBL1) and CCNB1 promoters. The authors model the simultaneous binding of LIN54-DBD and E2F4-DP2 to DNA and observe that they bind on opposite sides of DNA suggesting that this region would be nucleosome free. ENCODE micrococcal nuclease studies are consistent with this model and furthermore suggest that the region downstream of LIN54-DBD would contain a bound nucleosome thereby suggesting a model whereby the MuvB complex could repress gene expression by recruiting and maintaining an adjacent nucleosome. This is an interesting and novel model to explain the mode of repression by the MuvB complex.

The authors tested binding of transfected LIN54 variants to the p107 and CCNB1 promoters by ChIP (Fig. 5A). However, the p107 promoter is not known to harbor a CHR site, but contains an E2F DNA element. This information should be provided to the reader. According to the model of the authors (Fig. 6D) and consistent with E2F elements found to be enriched at DREAM bound regions (Ref 2), one would expect that binding of DREAM to p107 promoter is mediated by E2F4-DP interacting with the E2F element. Fig. 5A shows that binding of LIN54 to CCNB1 compared to p107 appears to depend more on LIN54's ability to bind a CHR. However, it is surprising that it is important for binding the p107 promoter at all. The authors should clarify this discrepancy and compare additional E2F element containing early cell cycle gene promoters, such as CDC6, E2F1, and MCM5 to CHR containing late cell cycle gene promoters, such as PLK1, CCNB2 and UBE2C. If the results are similar

to p107 and CCNB1, it should be discussed why LIN54 may be important to mediate DREAM binding to promoters not harboring a CHR.

Suggested improvements: experiments, data for possible revision

1. Figure 1A does not display the colors in the legend.
2. The abbreviation CHR stands for "cell cycle genes homology region" according to Zwicker et al. 1995 and Müller and Engeland 2010.
3. The gene name "CDC2" was changed to "CDK1" in the HUGO database some years ago.
4. Page 3-4. It would be helpful to the reader to explain why they switched from CHR27 to CHR13.
5. Page 12. The authors discuss TSO1, Tombola and CPP1 in the context of Fig. 3D. Tombola should be included in Fig. 3D.
6. Page 12. The authors cite "many of these promoters also contain a Myb binding site". The word "many" may be misleading when reference 14 finds up to 17%.
7. Page 13. The authors state they found "a few other examples in transcription factors". It would be helpful to the reader if some examples were given particularly where the Myb site was located in relationship to the TSS, CHR site and nucleosome binding region.
8. Figure 5B is missing input blots for Myb, V5 and Lin9.

References: appropriate credit to previous work?

1. Page 2. "LIN52 binds p130, LIN54 binds DNA, and RBAP48 binds histones" is not well supported by the References # 3, 8, 12. Ref. 7 better supports LIN52-p130 binding and Ref. 15 supports LIN54 DNA binding.
2. Page 8. appears have the wrong citation, "The average minor groove width [...] which is narrow but typical of A/T tracts." References 14 & 15 do not address minor grooves.
3. Page 10. The authors cite "the CDE serves as an E2F-DP binding site". However, the references only state that the CDE is related to E2F DNA elements suggesting that it can mediate binding of E2F-DP complexes but was never experimentally validated. Similarly for CHR too.
4. Page 11. In the discussion, the authors state "79% of DREAM bound late cell cycle promoters contain the CHR sequence defined by our study" and in the results they showed that the TT and AA at the edges of the CHR were critical. It would be helpful if the authors discussed functional CHR elements that bind DREAM but do not display the TTYRAA motif such as TTTGTA (Chek2), TTTGAT (Melk), TTTGAG (Pold1), TTCGAG (Rad18), and TTCGAT (Rad54l) as reported in Ref 14.

5. It would be helpful if the authors discussed, based on their structural insights, the phenomenon of a CHR site being sufficient to mediate binding of DREAM in some promoters, e.g. *Ccnb2* and *Ube2c* (Ref 17), while an additional CDE site being required in other promoters, e.g. *Plk4* (Fischer et al. 2014).

Reviewer #3 (Remarks to the Author)

The authors carry out a structural and functional study of the LIN54 DNA binding protein, a key component of the DREAM complex that drives a transcriptional program that regulates mammalian cell development and quiescence. They determine the crystal structure of the DNA binding domain of LIN54 bound to a consensus DNA defined by extensive binding studies. The structural and binding data reveal an interesting mechanism for sequence-specific DNA recognition involving Tyr residues that interact with the minor groove faces of AT base pairs. Mutagenesis confirms the importance and relevance of this complex for recruitment to cell cycle promoters. Overall the quality of the work is solid.

Major comments:

1. The structure is interesting in that the DNA consensus is 2-fold symmetric and the monomeric protein binds so that the N- and C- domains are oriented in a pseudo-symmetric manner on the DNA. The authors should consider whether it is possible that the DNA actually binds in two symmetric orientations related by a pseudo 2-fold positioned on the T-G/A-C step. Analysis of the unbiased electron density of the DNA at 2.4 Ang might possibly resolve this issue. If it is really true that the protein binds in a single orientation, can the authors explain why? i.e. why does the N domain bind one half-site, while the C-domain prefers the other? Does the binding data help to resolve this? Biologically this could be interesting if the protein needs to bind in a preferred single orientation to help organize the assembly of the larger protein complex, eg: in the context of a CDE site (Fig. 6). Does the modeling of the LIN54-E2F-DP complex suggest a protein-protein contact that might stabilize this interaction? Do these isolated domains cooperatively binding DNA?

2. The authors argue that conserved Tyr residues contact T-A bases in the minor groove across propeller twisted TT/AA steps. It would be interesting to see if CC/II substitutions could also be recognized with similar affinity. CC/GG should be disfavored due to a clash with N2 of G and reduced propensity for propeller twist. It is also interesting that the recognition of the propeller twisted bases seems to break the apparent H-bonding symmetry in the minor groove between A-T and T-A pairs. The authors might want to comment on this.

Minor points:

Fig. 1a - provide a legend to show which trace corresponds to which protein construct

Supp. Fig. 2, 3 - is the RMSD on all atoms or on CA only? Should be calculated on CA only.

Fig. 3. Indicate where the flexible linker should be and indicate this region on the sequence. Indicate that the spheres represent the bound Zn.

Supp. Fig. 3 - Show Zn atoms, would also be helpful to indicate how the Zn atoms are coordinated in each complex. Indicate N- and C-termini of chain. Might be helpful to show same orientation for each alignment.

We thank the reviewers for their insightful comments, and we have made the suggested improvements to the manuscript. The most significant changes in the revision include further structural analysis, a description of the sequences in the two promoters used in the chromatin immunoprecipitation experiment, and the addition of data that further explains the result that LIN54 mutation still influences its occupancy at the *p107* promoter, which contains a non-optimal CHR site. Important changes to the manuscript are highlighted in blue font.

Reviewer 1

Comment

1) It was stated in the manuscript that p130 and CCNB1 were chosen as examples because of their cell cycle-dependent expression. However, the particular CHR sites for the two genes are not presented. How do they correlate with the LIN54/CHR binding consensus? A CHR for CCNB1 has been published. Is there any reported CHR element for the p130 gene?

We agree that this important information was lacking in the text, and we have added explanation of the CDE and CHR sequences present in the *p107* and *CCNB1* promoters on Page 11. The *p107* promoter contains a CDE site and has a sub-optimal CHR site (TTTGAG), whereas the *CCNB1* promoter contains no CDE site and an optimal CHR site (TTTAAA).

Minor Comments

1) The title of the manuscript is too general. It should be more specific and include the abbreviated names 'LIN54' and 'CHR'.

We have changed the title as follows to add “LIN54”: “Structural Basis for Recognition of Cell Cycle Regulated Promoters by LIN54.” We believe that “Cell Cycle Regulated Promoters” is more descriptive than “CHR” and is therefore more suitable.

2) In the first paragraph of 'Discussion': '...we defined the CHR motif ... as TTYRAA...'. This should be worded differently as this consensus had been reported in Ref. 14, which has been referenced in several parts of the manuscript correctly.

We removed the words “we defined” from this sentence and kept our reference to the Muller et al. 2014 paper (now Ref. 17).

Reviewer 2

Major Comment

1) The authors tested binding of transfected LIN54 variants to the p107 and CCNB1 promoters by ChIP (Fig. 5A). However, the p107 promoter is not known to harbor a CHR site, but contains an E2F DNA element. This information should be provided to the reader. According to the model of the authors (Fig. 6D) and consistent with E2F elements found to be enriched at DREAM bound regions (Ref 2), one would expect that binding of DREAM to p107 promoter is mediated by E2F4-DP interacting with the E2F element. Fig. 5A shows that binding of LIN54 to CCNB1 compared to p107 appears to depend more on LIN54's ability to bind a CHR. However, it is surprising that it is important for binding the p107 promoter at all. The authors should clarify this discrepancy and compare additional E2F element containing early cell cycle gene promoters, such as CDC6, E2F1, and MCM5 to CHR containing late cell cycle gene promoters, such as PLK1, CCNB2 and UBE2C. If the results are similar to p107 and CCNB1, it should be discussed why LIN54 may be important to mediate DREAM binding to promoters not harboring a CHR.

As noted above in the first response to Reviewer 1, we now detail the CDE and CHR sequences in the two tested promoters. We somewhat agree that “it is surprising” the double tyrosine mutation in LIN54 decreases occupancy at the *p107* promoter. Although this promoter does not have a canonical CHR site, the promoter contains a weak CHR site (TTTGAG), which contains an intact half-site, and an additional CDE site. We propose that weak affinity to this poor CHR sequence, especially in the context of an adjacent CDE-binding site, may stabilize DREAM binding to the promoter. We offer this explanation now after describing the result on Page 11, and provide additional data to support it. We performed a similar ChIP experiment in HeLa cells, in which DREAM does not form, and found that mutation of the DNA-binding residues in LIN54 has little effect at the *p107* promoter. This result is consistent with the conclusion that binding of DREAM at the *p107* promoter is primarily driven by E2F binding the CDE element but has some contribution from LIN54 binding the suboptimal CHR element.

Suggested improvements

1) *Figure 1A does not display the colors in the legend.*

We corrected the legend to display the colors.

2) *The abbreviation CHR stands for "cell cycle genes homology region" according to Zwicker et al. 1995 and Müller and Engeland 2010.*

We corrected our definition of the CHR abbreviation as suggested in the abstract and introduction (Page 3).

3) *The gene name "CDC2" was changed to "CDK1" in the HUGO database some years ago.*

We changed the gene name to “CDK1” as suggested.

4) *Page 3-4. It would be helpful to the reader to explain why they switched from CHR27 to CHR13.*

We added the following explanation to the results section on Page 6: “We continued analyzing the shorter CHR13 sequence, because it was more suitable for structural studies.”

5) *Page 12. The authors discuss TSO1, Tombola and CPP1 in the context of Fig. 3D. Tombola should be included in Fig. 3D.*

Rather than include Tombola in Fig. 3d, we removed citing Tombola in the caption text. Tombola does not have two CxC domains like the other proteins in the alignment.

6) *Page 12. The authors cite "many of these promoters also contain a Myb binding site". The word "many" may be misleading when reference 14 finds up to 17%.*

We changed the word “many” to “some.”

7) *Page 13. The authors state they found "a few other examples in transcription factors". It would be helpful to the reader if some examples were given particularly where the Myb site was located in relationship to the TSS, CHR site and nucleosome binding region.*

The statement in the discussion references other rare examples of transcription factors structures that bind the minor groove using a tyrosine. Myb is not such an example, and the connection made by the reviewer here was unclear to us.

8) *Figure 5B is missing input blots for Myb, V5 and Lin9.*

We have added these input blots to the figure.

Comments about references

1) *Page 2. "LIN52 binds p130, LIN54 binds DNA, and RBAP48 binds histones" is not well supported by the References # 3, 8, 12. Ref. 7 better supports LIN52-p130 binding and Ref. 15 supports LIN54 DNA binding.*

We corrected the references as suggested.

2) *Page 8. appears have the wrong citation, "The average minor groove width [...] which is narrow but typical of A/T tracts." References 14 & 15 do not address minor grooves.*

We corrected the reference numbering, which had an error in the original manuscript.

3) *Page 10. The authors cite "the CDE serves as an E2F-DP binding site". However, the references only state that the CDE is related to E2F DNA elements suggesting that it can mediate binding of E2F-DP complexes but was never experimentally validated. Similarly for CHR too.*

We moved the references to the Engeland papers to before the semicolon in the sentence to clarify that they refer to the idea that DREAM binds promoters containing both CDE and CHR elements. We added a reference to the E2F-DP structure paper, which validates that E2F-DP binds to DNA sequences that match the CDE element.

4) *Page 11. In the discussion, the authors state "79% of DREAM bound late cell cycle promoters contain the CHR sequence defined by our study" and in the results they showed that the TT and AA at the edges of the CHR were critical. It would be helpful if the authors discussed functional CHR elements that bind DREAM but do not display the TTYRAA motif such as TTTGTA (Chek2), TTTGAT (Melk), TTTGAG (Pold1), TTCGAG (Rad18), and TTCGAT (Rad54l) as reported in Ref 14.*

We address this suggestion with the following statement in the discussion:

Our ChIP data suggest that DREAM binding to the p107 promoter, which contains a noncanonical CHR site, is still influenced by LIN54-DNA interactions. The p107 promoter (TTTGAG) and other promoters such as MELK (TTTGAT), RAD18 (TTCGAG), and RAD54L (TTCGAT), which bind DREAM but lack a canonical CHR¹⁷, have at least half of the consensus sequence intact and contain a proximal CDE site. We suggest that DREAM binding at these promoters is mediated by the E2F-CDE binding and additional, weak interactions between LIN54 and the noncanonical CHR.

We note here that we found in Ref. 17 that POLD1 and CHEK2 do contain canonical CHRs within the promoter or gene.

5) *It would be helpful if the authors discussed, based on their structural insights, the phenomenon of a CHR site being sufficient to mediate binding of DREAM in some promoters, e.g. Ccnb2 and Ube2c (Ref 17), while an additional CDE site being required in other promoters, e.g. Plk4 (Fischer et al. 2014).*

We believe that this point is addressed by the sentences we added to the discussion in response to point 4 above. We propose that promoters with non-canonical CHR sites may also require CDE sites for DREAM binding.

Reviewer 3

Major Comments

1) *The structure is interesting in that the DNA consensus is 2-fold symmetric and the monomeric protein binds so that the N- and C- domains are oriented in a pseudo-symmetric manner on the DNA. The authors should consider whether it is possible that the DNA actually binds in two symmetric orientations related by a pseudo 2-fold positioned on the T-G/A-C step. Analysis of the unbiased electron density of the DNA at 2.4 Ang might possibly resolve this issue. If it is really true that the protein binds in a single orientation, can the authors explain why? i.e. why does the N domain bind one half-site, while the C-domain prefers the other? Does the binding data help to resolve this? Biologically this could be interesting if the protein needs to bind in a preferred single orientation to help organize the assembly of the larger protein complex, eg: in the context of a CDE site (Fig. 6). Does the modeling of the LIN54-E2F-DP complex suggest a protein-protein contact that might stabilize this interaction? Do these isolated domains cooperatively binding DNA?*

The reviewer poses a number of interesting questions, some of which we can address. As suggested, we examined the unbiased electron density to confirm the orientation of the DNA. We find that in the crystal structure, the DNA is only in one orientation and the two DNA-binding domains bind the half-sites with specificity. We added this point to the results on Page 8, and we added Supplementary Fig. 5, which shows an omit map of the asymmetric bases in the sequence. We cannot be sure with our current binding data whether the same specificity exists in solution. We now note in the discussion that our NMR data indicates a higher affinity interaction between DBD-C and the DNA and that this observation is consistent with the 5'-TTT half of the consensus being more conserved in DREAM binding sites. Although we are pursuing structural studies of the entire complex, we cannot be sure at this time whether the particular orientation is important for assembly and orientation; however, we raise this possibility in the discussion on Page 15.

2) *The authors argue that conserved Tyr residues contact T-A bases in the minor groove across propeller twisted TT/AA steps. It would be interesting to see if CC/II substitutions could also be recognized with similar affinity. CC/GG should be disfavored due to a clash with N2 of G and reduced propensity for propeller twist. It is also interesting that the recognition of the propeller twisted bases seems to break the apparent H-bonding symmetry in the minor groove between A-T and T-A pairs. The authors might want to comment on this.*

We performed the proposed experiment and found that LIN54 binds CHR13 sequences containing CC/II substitutions with intermediate affinity between TT/AA (relatively high affinity) and CC/GG (no affinity) containing sequences. As suggested by the reviewer, these data support the hypothesis that recognition of deformed base steps is critical, and we include these data in Supplementary Fig. 7.

Minor Comments

1) *Fig. 1a - provide a legend to show which trace corresponds to which protein construct*

We have added the appropriate legend to Fig. 1A.

2) *Supp. Fig. 2, 3 - is the RMSD on all atoms or on CA only? Should be calculated on CA only.*

We have now indicated that the RMSD was calculated for C-alpha atoms only.

3) Fig. 3. Indicate where the flexible linker should be and indicate this region on the sequence. Indicate that the spheres represent the bound Zn.

We clarified the figure and legend as suggested.

4) Supp. Fig. 3 - Show Zn atoms, would also be helpful to indicate how the Zn atoms are coordinated in each complex. Indicate N- and C-termini of chain. Might be helpful to show same orientation for each alignment.

We made all these suggested changes to Supplementary Fig. 3.

Reviewer #1 (Remarks to the Author):

1) The title of the manuscript is too general. It should be more specific and include the abbreviated names 'LIN54' and 'CHR'.

We have changed the title as follows to add "LIN54": "Structural Basis for Recognition of Cell Cycle Regulated Promoters by LIN54." We believe that "Cell Cycle Regulated Promoters" is more descriptive than "CHR" and is therefore more suitable.

No. "Cell Cycle Regulated Promoters" is still too general. There are many more promoters regulated during the cell cycle than there are CHR-controlled promoters, e.g. the large group of E2F genes. My suggestion is: "Structural Basis for Recognition of Cell Cycle Regulated Promoters by LIN54 binding CHR Elements."

Reviewer #2 (Remarks to the Author):

The manuscript has been significantly improved in clarity and accuracy by the changes made in response to the critiques. In particular, the additional insight provided into how the non-canonical CHR site in p107 contributes to DREAM binding in T98G cells but not HeLa cells is an important addition. The electron density map and supplementary figure 7 are helpful additions as well.

Reviewer #3 (Remarks to the Author):

The authors have adequately addressed my comments and I recommend publication.

Reviewer 1

Comment

No. "Cell Cycle Regulated Promoters" is still too general. There are many more promoters regulated during the cell cycle than there are CHR-controlled promoters, e.g. the large group of E2F genes. My suggestion is: "Structural Basis for Recognition of Cell Cycle Regulated Promoters by LIN54 binding CHR Elements."

We changed the title to "Structural Basis for LIN54 Recognition of CHR Elements in Cell Cycle Regulated Promoters." We believe this preferred title still captures the specificity of "CHR Elements" as suggested.